# Development and testing of an opioid tapering self-management intervention for chronic pain: I-WOTCH

Harbinder Kaur Sandhu ![ORCID],[1] Jane Shaw,[2] Dawn Carnes,[3] Andrea D Furlan ![ORCID],[4,5] Colin Tysall,[6,7] Henry Adjei,[1] Chockalingam Muthiah,[1] Jennifer Noyes,[2] Nicole K Y Tang ![ORCID],[8] Stephanie JC Taylor ![ORCID],[3] Martin Underwood ![ORCID],[1,9] Adrian Willis,[1] Sam Eldabe,[2] On behalf of the I-WOTCH Team

For numbered affiliations see end of article.

**Correspondence to**
Professor Harbinder Kaur Sandhu;
harbinder.k.sandhu@warwick.ac.uk

## ABSTRACT

**Objectives** To describe the design, development and pilot of a multicomponent intervention aimed at supporting withdrawal of opioids for people with chronic non-malignant pain for future evaluation in the Improving the Wellbeing of people with Opioid Treated CHronic pain (I-WOTCH) randomised controlled trial.

**Design** The I-WOTCH intervention draws on previous literature and collaboration with stakeholders (patient and public involvement). Intervention mapping and development activities of Behaviour Change Taxonomy are described.

**Setting** The intervention development was conducted by a multidisciplinary team with clinical, academic and service user perspectives. The team had expertise in the development and testing of complex health behaviour interventions, opioid tapering and pain management in primary and secondary care, I.T programming, and software development—to develop an opioid tapering App.

**Participants** The I-WOTCH trial participants are adults (18 years and over) with chronic non-malignant pain using strong opioids for at least 3 months and on most days in the preceding month.

**Outcomes** A multicomponent self-management support package to help people using opioids for chronic non-malignant pain reduce opioid use.

**Interventions and results** Receiving information on the impact of long-term opioid use, and potential adverse effects were highlighted as important facilitators in making the decision to reduce opioids. Case studies of those who have successfully stopped taking opioids were also favoured as a facilitator to reduce opioid use. Barriers included the need for a 'trade-off to fill the deficit of the effect of the drug'. The final I-WOTCH intervention consists of an 8–10 week programme incorporating: education; problem-solving; motivation; group and one to one tailored planning; reflection and monitoring. A detailed facilitator manual was developed to promote consistent delivery of the intervention across the UK.

**Conclusions** We describe the development of an opioid reduction intervention package suitable for testing in the I-WOTCH randomised controlled trial.

**Trial registration number** ISRCTN49470934.

## Strengths and limitations of this study

► The Improving the Wellbeing of people with Opioid Treated CHronic pain (I-WOTCH) Intervention draws on psychological and behaviour change frameworks.
► The I-WOTCH intervention was developed with key stakeholders including patient and public involvement (those with chronic-non-malignant pain and experience of opioid use and/or tapering).
► The pilot phases and feasibility testing gave valuable feedback and changes were made to the intervention accordingly.
► At the time of designing the intervention, there was limited previous work and information to inform content of the intervention.

## INTRODUCTION

Pain, and pain related disorders, continue to be the leading cause of disability and disease burden globally,[1] with low back pain making the largest contribution to years lived with disability. In England, at least 8 million people (15% of the population) have moderate-to-severe persistent (chronic) pain[2] defined as *pain that lasts or recurs for more than 3 months.*[3] Over the past few decades, there has been a global epidemic of opioid prescribing for chronic non-malignant pain. A 2020 systematic review found that 30% of people with chronic non-malignant pain are prescribed opioid medication and, globally, this has steadily increased until recently with time.[4] In the UK, prescribing rates have decreased slightly over recent years; however, the number of prescriptions remains high.[5] Long-term use of opioids leads to tolerance and loss of effective pain relief. Adverse consequences include opioid-induced hyperalgesia, endocrine hypogonadism, drowsiness, a high risk of dependency, opioid use disorder, sleep apnoea, immune suppression, falls leading to increased fractures

(particularly a risk in the elderly population) and increased risk for overdose and death.[6] There are limited strategies to help with risk mitigation and evidence based interventions to help people with chronic non-malignant pain withdraw from opioids.[7] A 2020 systematic review found ten randomised controlled trials (n=835) of patient-focused opioid de-prescribing interventions targeting people with chronic non-malignant pain. These included: dose reduction protocols (weekly reduction of 10%); opioid replacement including (buprenorphine, morphine sulphate or oxycodone hydrochloride or varenicline; non-pharmacological therapies including mindfulness (vs active control or support group); therapeutic interactive voice response programme (vs usual care); meditation; cognitive–behavioural therapy (vs usual care); and electroacupuncture (vs sham). The primary outcome was mean reduction of daily dose in morphine milligram equivalents (MME). Only one study reported a statistically significant difference in the daily dose between groups in favour of the intervention (a study using a dose tapering protocol) (mean difference −27.9 MME/day, 95% CI −41.1 to −14.7).[8] None of these interventions reported increases in opioid cessation in the intervention groups. Overall, the authors were unable to recommend any particular deprescribing strategy due to the small number of studies and heterogeneity of the data.[4]

Current recommendations on opioid tapering are based on the best practice and guidelines which need to be supported by further evidence.[9] Here, we describe the development of a multicomponent opioid tapering programme (incorporating group and one to one sessions) as part of the I-WOTCH study (Improving the Wellbeing of people with Opioid Treated CHronic pain), funded by the National Institute of Health Research (14/224/04). The I-WOTCH study protocol has been published previously.[10]

## METHODS

The I-WOTCH intervention was developed in collaboration with the target population (those with chronic non-malignant pain and experience of opioid use). It employed theory and evidence-based implementation (with a view to implementation in the real world should it be effective) and included digital technologies to generate opioid tapering plans.[11] The Medical Research Council Framework[12] for designing complex interventions, evidence-based interventions[13] and core theoretical principles were used to inform the design of content, structure and delivery of the intervention.

Key stages of the intervention development are shown in figure 1. Adjustment and adaptation to the intervention were implemented inline with feedback received from stakeholders (service users, clinicians and facilitators delivering the I-WOTCH intervention).

### Aims and objectives of the I-WOTCH intervention

In line with the overall study, the aims of the I-WOTCH intervention were:

1. To reduce opioid and healthcare use for people with chronic, non-malignant pain.
2. To increase study participants' self-efficacy (confidence) to reduce opioid medication and implement self-management strategies of pain.
3. To improve quality of life and help people live better with pain.

Objectives:

1. To provide education using a range of teaching methods; group discussion, problem-solving, experiential learning and case studies.
2. To provide an environment which enhances motivation to reduce opioid use through group cohesion and one to one support.
3. To provide an overall cost-effective intervention to be implemented in healthcare services.

Defining the aims and objectives enabled us to consider what we wanted to achieve, how and for what purpose. In addition, we were aware of potential facilitators and barriers that could influence engagement with the intervention and the procedures of the trial. Figure 2 shows the direction of travel we were aiming for and what we needed to consider when designing the detail of the intervention and mechanism of behaviour change.

### Patient and public involvement

During the development stages of I-WOTCH, we held two PPI meetings with the Clinical Research Network (North East and Cumbria) at The James Cook University Hospital (South Tees Hospitals NHS Foundation Trust). A total of 19 volunteer participants (people with chronic pain and experience of opioid therapy and/or opioid tapering) attended. Discussions were facilitated by members of the study team (HS, DC, JS and SE) and included, intervention structure and design, content (topics to cover which would potentially increase motivation and confidence to taper opioids), length of programme, where the intervention should be delivered, support during opioid tapering (including frequency of contact with healthcare professionals) and delivery of the intervention (who should deliver the intervention) (table 1). In addition to this, two lay advisors who were part of the I-WOTCH study recruited via Universities/User Teaching[14] gave considerable input into the design of, and training to deliver, the intervention.

### Opioid tapering and behaviour change

The target behaviour change was defined as the participants engaging in the I-WOTCH intervention: reducing participant opioid use, and implementing non-pharmacological strategies of pain management. The biopsychosocial framework,[15] Michie's taxonomy of behaviour change and the COM-B framework for behaviour change (Capability, Opportunity, Motivation) were consulted.[16] Capability includes psychological capability (eg, can patients engage

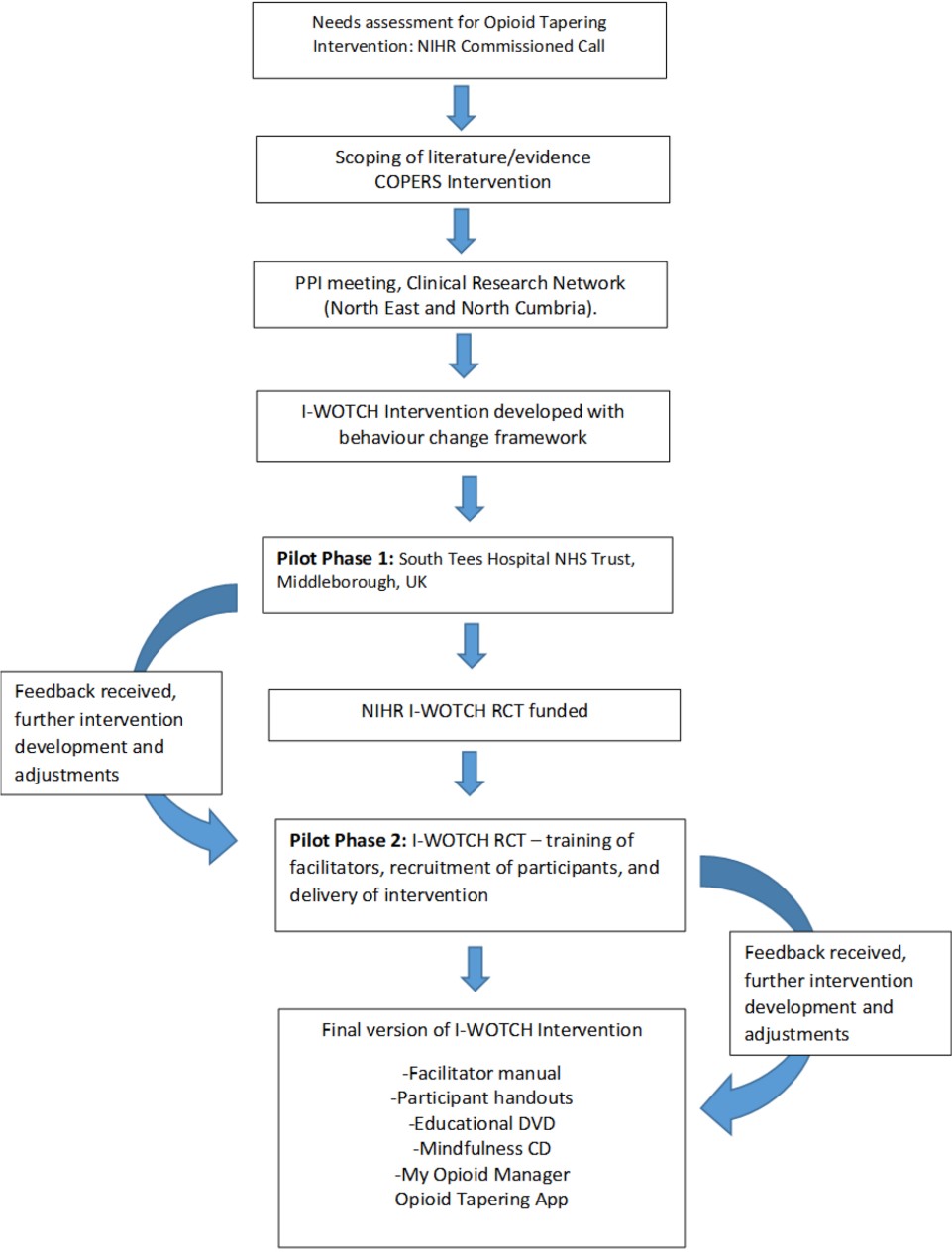

**Figure 1** Stages of Improving the Wellbeing of people with Opioid Treated CHronic pain (I-WOTCH) intervention development. NIHR, National Institute for Health Research; RCT, randomised controlled trial.

in the necessary thought processes needed to commit and adhere to the tapering processes?) and physical capability (eg, do participants have the capacity to engage in the tapering?). Psychological capability is broken down to cognitive functioning and executive functioning. To promote cognitive functioning (which includes a range of mental abilities such as learning, problem-solving and attention), we produced handouts of material covered on each day of the programme. This allowed opportunity for participants to recap over the core messages and information in their own time. We also included time for group reflection at the start of each session and summarising discussions at the end of each of the group days (with opportunities for questions). In addition to this, we developed an educational DVD, a mindfulness

CD and relaxation CD for each participant (at the time we developed the intervention DVDs and CDs were still in common use). By providing material to take home, we were giving participants an opportunity to revisit and take in the information at their own pace.[17] Executive functioning includes the capacity to plan and think, explore challenges that may occur (eg, fear of withdrawal symptoms), stay focused on the goal (opioid reduction) and resist temptation.[18] In the I-WOTCH intervention, we gave participants opportunity to set goals (through an educational session and support in generating goals related to opioid tapering and their general life). We also encouraged self-reflection to identify perceived barriers and facilitators to tapering and gave further guidance to overcome the perceived barriers in the tailored one to

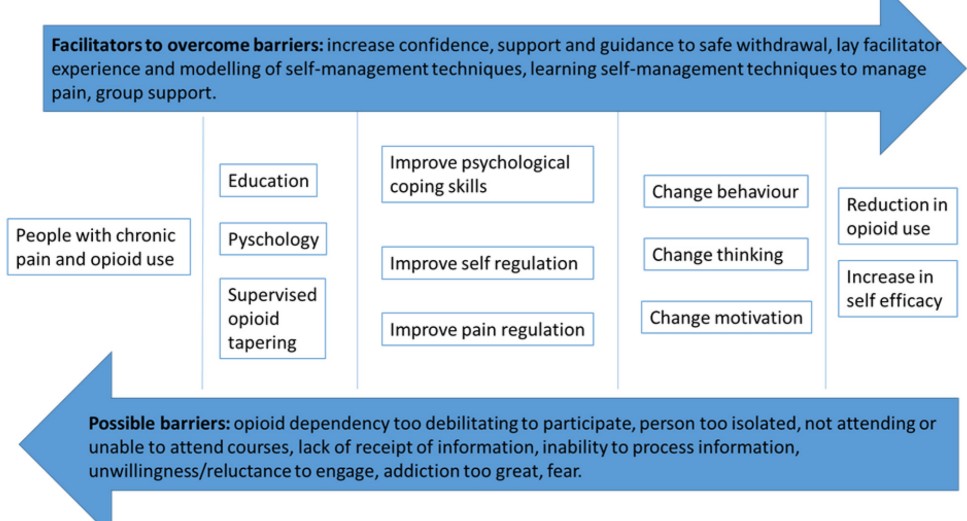

**Facilitators to overcome barriers:** increase confidence, support and guidance to safe withdrawal, lay facilitator experience and modelling of self-management techniques, learning self-management techniques to manage pain, group support.

| People with chronic pain and opioid use |
| Education |
| Pyschology |
| Supervised opioid tapering |
| Improve psychological coping skills |
| Improve self regulation |
| Improve pain regulation |
| Change behaviour |
| Change thinking |
| Change motivation |
| Reduction in opioid use |
| Increase in self efficacy |

**Possible barriers:** opioid dependency too debilitating to participate, person too isolated, not attending or unable to attend courses, lack of receipt of information, inability to process information, unwillingness/reluctance to engage, addiction too great, fear.

**Figure 2** Reducing opioids for people with chronic non-malignant pain.

one support sessions with the clinical facilitator. Physical capability refers to whether the participants exposed to the I-WOTCH intervention felt they had the right skills to engage in the tapering of their opioids, this may include management of withdrawal, confidence and having structure and support in place. The I-WOTCH intervention was designed to help participants adapt and put into place lifestyle changes.

Opportunity is the second component of the COM-B model. For this, we explored factors external to the individual that would promote opioid tapering. For example, physical opportunity includes costs of opioids and

| Table 1 | Feedback from PPI informing intervention development |
|---|---|
| **Discussion topic** | **Feedback informing intervention development** |
| Behaviour change | Agreed aims should be a reduction in opioid consumption and engagement in the I-WOTCH programme. Behaviour change needs to be accepted before opioid reduction can occur. |
| Understanding motivation to change behaviour | Changing medication and reducing medication can be motivated by: (i) a trade-off to fill the deficit of the effect of the drug (something else needed that is as effective as the drug they would lose) (ii) reduction in side effects Use of case studies of people who had successfully stopped taking opioids would be useful. |
| Content and topics to be covered | The intervention would benefit from being informative (opioid education, especially long-term consequences, pros and cons of opioid use and managing withdrawal). The following topics were recommended for inclusion: ► What is pain ► Acceptance—pain and learning to live better with pain ► Impact of pain – and integrate this information with taking medication (Opioids), why and how? ► The importance of hobbies and having a distraction to manage the pain ► Offer alternative non-pharmacological ways of coping, for example, mindfulness and relaxation ► Incorporate movement ► Guidance on posture and exercise/activity ► Pacing—not over doing things |
| Dependency versus addiction | It was felt important to distinguish between dependency and addiction, as some were concerned about the stigma and labels attached to long-term opioid use for chronic pain. |
| Delivery of I-WOTCH Intervention, who? | Feedback favoured the course to be delivered jointly by a HCP and a lay facilitator (someone who had experience of long-term pain and opioid use/tapering). |
| Structure of intervention | Group and individual care approaches were valued. Length of the proposed programme (3-day group sessions and ongoing one to one support) was supported. The duration of intervention was not viewed as burdensome given that some had people who had experienced severe withdrawal symptoms, and therefore ongoing support over the 8–10 weeks was needed. There was a consensus that a group-based format and group cohesion would be optimal because of the potential for social comparison, social validation and development of social support. Volunteers identified the impact of opioid use on enhanced day-to-day activities as important evaluation outcomes, including: work productivity, looking after children, and overall functioning. |
| Communication during study | Volunteers welcomed the idea of having a study website to give participants an opportunity to be updated about the study as a whole and progress. |

HCP, healthcare professional; I-WOTCH, Improving the Wellbeing of people with Opioid Treated CHronic pain.

travel, access and availability, developing a tapering plan (clear and informative) and enhancing communication between the clinical facilitator and participant through motivational interviewing (MI) during the tapering processes. In relation to social opportunity, we referred to what other factors may impact the decision to taper such as stigma and cultural beliefs.

Motivation, this refers to both the cognitive motivation and emotional processes to energise and direct the behaviour change of opioid tapering. Reflective processes included exploring perceptions and meaning of chronic pain during the group sessions as well as beliefs about tapering, possible outcomes concerns and self-efficacy. There was opportunity to evaluate and be reflective during the group sessions as well as one to one support. Automatic processes refer to the emotional responses which may occur during the I-WOTCH intervention and these include anxiety, fear, stress and low mood. All topics were covered in the group sessions including recognition of thoughts and emotions and management strategies.

Each component of the I-WOTCH intervention was informed and mapped on to behaviour change taxonomies. The intervention also drew on psychological theories of self-efficacy,[19] theory of planned behaviour and reasoned action,[20 21] social learning[22] and group-based interventions,[23] cognitive–behavioural change,[24] MI[25] and evidence-based interventions for self-managing chronic pain (COPERS)[26] described in table 2.

## Feasibility testing

Funding from the Hambelton and Richmond Clinical Commissioning Group for a community pain management service allowed us to test the feasibility of the I-WOTCH intervention. Seven people were trained by the study team to deliver the intervention (three community team clinicians, two nurses and two volunteer patients). Two courses were observed by a member of the study team to evaluate how the course content was delivered and received by both the group facilitators and the group participants (five participants in total). Discussions included, what worked well, what did not work well, and whether participants felt that the aims and objectives of the programme were met and suggestions for changes.

The second stage of feasibility was part of the pilot phase of the randomised controlled trial and involved facilitator training for the trial. Two groups were delivered in Coventry. From both stages of feasibility testing, feedback was taken on board and adaptions implemented for the training (table 3) and course content and structure (table 4).

Overall, the feedback regarding the content of programme was positive. Participants felt that the distraction techniques worked well and helped break up the sessions. They also valued understanding the link between mood and pain and found the case studies useful in helping to motivate them to start reducing their opioids. Facilitators and participants in both pilot phases reported that it was an informative interactive course. Observations showed good delivery and interaction between facilitators and participants, good use of questions and answer sessions and role play. Both facilitators and patients agreed it may have been more interactive had the group been larger.

## Final I-WOTCH intervention

The final I-WOTCH intervention (figure 3) consists of group day 1 (delivered week 1), group day 2 (delivered week 2), a one-to-one consultation with an I-WOTCH trained nurse (also in week 2 and after group day 2), group day 3 (week 3) and then two telephone consultations and a final face-to-face consultation to offer continual support for tapering. Each component of the intervention builds on previous knowledge and experience, and where the one-to-one consultation allows consolidation and tailoring of advice and support for tapering. At the beginning of the intervention, the learning is centred on pain and opioid education, with day 2 of the programme then introducing changes in beliefs and adapting different strategies as reduction of opioids occur. It is at this point tailoring support and MI to action a change in beliefs is promoted through the one-to-one support sessions with an opportunity for further regulation and group cohesion/support on the wider impact of opioid reduction and long term behaviour change (group day 3). The additonal one to one support, self-regulation, reflection and monitoring.

## One-to-one consultations

The one-to-one sessions with a trained I-WOTCH nurse were based on a MI model.[27] The aims of MI are to enhance behaviour change through a patient-centred framework, where the patient is able to explore personal goals, ambivalence to change and reach self-actualisation in a supportive environment. We trained the I-WOTCH nurses on the five principles of MI: (1) expressing empathy through reflective learning, (2) expressing empathy through reflective listening, (3) developing discrepancy between participant goals or values (related to opioid tapering and pain management) and their current behaviour, avoiding argument and direct confrontation, (4) adjusting to participant resistance to reducing opioid reduction rather than opposing it directly and (5) supporting self-efficacy and optimism. The one-to-one consultations included a review of medication, reflection on the opioid education and group session where case studies and information were presented and exploring any challenges to opioid tapering such as concerns about withdrawal. Nurses were also trained to calculate total opioid daily dose and how to use that to produce a tapering regime according to the I-WOTCH study protocol. Although MI has been widely applied in substance misuse, there are limited data available for its use in opioid cessation for people with chronic non-malignant pain. A 2020 pilot study testing MI to support opioid tapering in post joint arthroplasty surgery found a 62% increase in the rate of participants returning to baseline opioid use after surgery (HR 1.62; 95% CI 1.06 to 2.46; p=0.03).[28] Opioid tapering conversations maybe

**Table 2** Behaviour change taxonomy and opioid tapering

| I-WOTCH group based sessions day 1 (week 1) | Aims | Theoretical underpinnings | Behaviour change taxonomy |
|---|---|---|---|
| Introductions, group work, aims | To allow participants to introduce themselves to the group, encourage participation in a safe and relaxed environment, explore expectations and discuss the I-WOTCH course aims | Social cognitive theory<br>Biopsychosocial theory | Improve bonding and group cohesion.<br>Breaking barriers and encouraging self and social awareness |
| What causes pain? (pain information) | To increase understanding about long-term pain | Biopsychosocial theory<br>Principles of self-efficacy and acceptance | Credible source |
| Living with pain (Opioid education I) | To increase understanding about use of opioids for long-term pain and encourage participants to start questioning their own knowledge and beliefs about opioids and why they take them | Biopsychosocial theory<br>Theory of planned behaviour and reasoned action<br>Health beliefs | Information about health consequences |
| Acceptance | To understand and start to accept pain, with a view to implementing self-management strategies as reduction of opioids occurs | Acceptance and self-management of chronic pain | Goal setting<br>Commitment |
| Attention control and distraction | To learn how to focus the mind away from pain thoughts and use of opioids | Cognitive–behavioural change<br>Self-management of chronic pain<br>Health beliefs | Distraction |
| Distraction activity—drawing | An opportunity to practise distraction activity and socially interact with group informally | Cognitive–behavioural change<br>Social learning | Behavioural practice<br>Distraction |
| Good days, bad days—pain, bearable or not? | To reinforce that pain is not just physiological, it is a psychological, social and an emotional phenomenon | Biopsychosocial theory<br>Health beliefs | Information and antecedents<br>Information about health consequences<br>Reattribution of behaviour |
| The pain cycle (including opioids) and breaking the pain cycle | To explain and identify unhelpful factors in the pain cycle and learn strategies to break the cycle | Biopsychosocial theory<br>Health beliefs | Behaviour substitution (adding in other behaviours to break cycle) |
| Posture and movement | To promote body awareness, posture and muscle weakness (managing pain without opioids) | Theory of planned behaviour and reasoned action | Guidelines on exercise, physical therapy principles<br>Mindfulness |
| Relaxation and breathing | To reduce muscle tension and introduce breathing as a relaxation technique | Cognitive—behavioural change<br>Self-management of chronic pain | Behavioural practice<br>Distraction<br>Body changes |
| Summary of the day | To consolidate learning of the day and outline aims for group day 2. | Acceptance and principles of self-efficacy | Action planning<br>Verbal persuasion about capability |
| **I-WOTCH group-based Sessions Day 2 (week 2)** | **Aims** | **Theoretical underpinnings** | **Behaviour change taxonomy** |
| Reflections from day 1 | To understand and empathise with the group | Social learning<br>Self-efficacy | Improve bonding and group cohesion, social cognitive theory |
| Stress-busting for Health: Action planning, problem-solving, pacing, SMART goal setting | To help the participants logically and systematically identify problems, free think solutions, set achievable goals and create action plans, as a means of escaping the pain cycle | Cognitive–behavioural change<br>Theory of planned behaviour and reasoned action | Goal setting<br>Comparative imagining of future outcomes<br>Reduce negative emotions<br>Problem-solving |
| Withdrawal symptoms, case studies (Opioid education II) | To discuss potential withdrawal symptoms that participants might experience if their taper is too quick | Health beliefs<br>Social learning | Social comparison (drawing attention to others' performance to allow comparison with the person's own performance)<br>Credible source<br>Comparative imagining of future outcomes |
| Distraction activity—origami | To learn how to focus the mind away from pain thoughts and use of opioids | Cognitive–behavioural change<br>Social learning | Behavioural practice<br>Distraction |

Continued

**Table 2** Continued

| I-WOTCH group-based Sessions Day 2 (week 2) | Aims | Theoretical underpinnings | Behaviour change taxonomy |
|---|---|---|---|
| Identifying and overcoming barriers to change | Introduce ideas about unhelpful thoughts, automatic thoughts and errors in thinking. To identify reasons why people stay in the pain cycle, and barriers to change. Introduce positive reframing | Cognitive–behavioural change Self-management of pain | Problem-solving Reduce negative emotions Framing/reframing |
| Mindful attention control | To introduce Mindfulness as a tool to train attention and distract from pain | Principles of mind body therapies and biofeedback and visualisation | Behavioural practice Distraction Body changes |
| Balance and stretch | To promote body awareness and core strength | Guidelines on exercise Physical therapy principles | Demonstration of behaviour Behavioural practice |
| Summary of the day | To consolidate learning of the day and outline aims for final group day 3. A reminder to attend the one to one appointment with the clinical facilitator. | Acceptance and principles of self-efficacy | Action planning Verbal persuasion about capability |
| **I-WOTCH group based Sessions day 3 (week 3)** | **Aims** | **Theoretical underpinnings** | **Behaviour change taxonomy** |
| Reflections from day two | To understand and empathise with the group and ascertain current thoughts | Social learning Self-efficacy | Review of behaviour |
| Anger, irritability and frustration | Identifying reasons for negative emotions and implementing goal setting and action planning | Cognitive–behavioural change Theory of planned behaviour and reasoned action | Reduce negative emotions Goal setting Action planning |
| Relationships: getting the most from your healthcare team (part1) | To reflect on consulting behaviour and promote effective communication and constructive consultations | Biopsychosocial theory Theory of planned behaviour and reasoned action | Information about antecedents Instruction on how to perform a behaviour (communication skills) |
| Relationships (part 2) listening skills | To improve listening and communication skills | Biopsychosocial theory Theory of planned behaviour and reasoned action | Social support (emotional) |
| Managing setbacks and non-drug management techniques | To know what to do when experiencing a setback or a flare up | Cognitive–behavioural change Self-efficacy | Anticipated regret Focus on past success |
| Mindful distraction activity – colouring | To learn how to focus the mind away from pain thoughts and use of opioids | Principles of mind body therapies and biofeedback and visualisation | Behavioural practice Distraction Body changes |
| Stretch | To learn how to stretch muscles gently with low risk of injury and pain | Biopsychosocial theory Self-efficacy Principles of acceptance | Demonstration of behaviour Behavioural practice |
| Mindfulness of thoughts and senses | To learn how to apply mindfulness of thoughts by detaching emotion from reality, to appreciate 'the now' | Principles of mind body therapies Biofeedback and visualisation | Distraction |
| Summary of the day | To consolidate the days learning. | Acceptance and principles of self-efficacy | Action planning |
| Summary of the course | To clarify learning from past three group days and motivation to continue with opioid reduction | Acceptance and principles of self-efficacy | Review of behaviour Verbal persuasion about capability |
| **One to one session** | **Aim** | **Theoretical Underpinnings** | **Behaviour Change Taxonomy** |
| Interaction one: face to face with clinical facilitator | To reflect on group learning days, agree tapering goals and generate tapering plan | Cognitive–behavioural change Motivational Interviewing | Goal setting behaviour Action planning Graded task Pros and cons |
| Interaction two: 30 min via telephone call with clinical facilitator | To reflect on progress and offer support during the tapering process | Cognitive–behavioural change Motivational Interviewing | Review behaviour Behavioural contract (adapted – as generated plan written) Social reward (congratulating on effort made and progress towards tapering-verbal) |

Continued

**Table 2**  Continued

| One to one session | Aim | Theoretical Underpinnings | Behaviour Change Taxonomy |
|---|---|---|---|
| Interaction three: 30 min via telephone with clinical facilitator | To reflect on progress and offer support during the tapering process | Cognitive–behavioural change Motivational Interviewing | Identification of self as role model (their own behaviour may be an example to others as they taper) |
| Interaction four: face to face with clinical facilitator | To reflect on progress so far and discuss goals for future | Cognitive–behavioural change Motivational Interviewing | Review behaviour Review outcome goal If applicable: discrepancy between current behaviour and goal feedback on behaviour Goal setting (behaviour) Goal setting (outcome) Action planning |

challenging and each participant will bring their own experiences and motivation to change; however by using MI as a tool, we encouraged I-WOTCH facilitators to support participants in their tapering.[29]

### One-to-one tapering: App

We adopted an opioid tapering regimen based on the Mayo Clinic experience as it provided some evidence to support the notion that slow tapering is unlikely to be associated with severe withdrawal symptoms and therefore likely to facilitate adherence.[30] This consisted of a 10% reduction of the original total daily dose every 7 days until a 30% of the original daily dose is reached. This is followed by a weekly decrease by 10% of the remaining dose. The 10% was rounded up to suit prescribing. For the calculation of equi-analgesic doses, we used the tables from the Faculty of Pain Medicine.[31] In order to ensure standardisation of tapering methodology across sites and various opioid preparations,

**Table 3**  Feedback and changes pilot phases I and II: training

| Feedback (pilot phase I and II)—training and facilitator feedback | Changes implemented |
|---|---|
| Facilitators agreed it is useful to go through the manual step by step, to gain familiarity with each component and navigate through the different stages. They preferred this rather than going through generic topics. | We incorporated this information into the training and prior to a group being delivered, if needed the study team helped to arrange meetings between the facilitators. |
| Facilitators felt it would be useful for all material to be emailed prior to the training to allow time for familiarisation with the manual. | Throughout the I-WOTCH study all course materials were sent to facilitators prior to training. |
| Facilitators suggested that during the training it would be useful to actually practice some of the sessions. | Where possible during the training days we incorporated case studies and role play, as well as experiential learning of mindfulness and using the tapering app to calculate opioid reduction doses. |
| Facilitators suggested that it would be useful if the course slides were numbered in correspondence to the sections in the manual. | All course slides were numbered and added to the manual for reference. |
| Facilitators also suggested that it would be useful to include the rationale for each topic into the manual, as it helped with their understanding of each topic and with their explanation to participants. | The rationale for each topic was included in the manual. |

I-WOTCH, Improving the Wellbeing of people with Opioid Treated CHronic pain.

**Table 4**  Feedback and changes pilot phases I and II: course content and structure

| Feedback (pilot phase I and II) participant feedback | Changes implemented |
|---|---|
| During pilot phase I, feedback favoured spreading the group sessions over 3 weeks (one group day per week). This was to help with consolidation of information and learning between sessions and also felt less burdensome. | In the I-WOTCH study, groups were delivered with this format (every Monday where possible for 3 weeks). |
| It was suggested the balance session worked well after the session on posture, to allow more understanding and connection with body. | This was changed in the I-WOTCH programme: balance and stretch was introduced on day 2 of the programme and posture and movement on day 1 of the programme. |
| Day 1 presented a lot of educational information on opioids and it was suggested to split this over 2 days to help support consolidation of understanding | The educational information was split over 2 days (day 1 and day 2 of the programme). |
| It was also suggested to move the session on pacing to after the pain cycle has been discussed, to help with the understanding of why pacing is important and can help break the unhelpful cycle. | The pain cycle was introduced and on day 1 of the programme and pacing was moved to day two of the programme. |
| During pilot phase I, patients welcomed an educational DVD to help with the learning. | As part of the I-WOTCH study, we produced an I-WOTCH education DVD which is used in the delivery of the programme, participants are able to then take this home and watch with their family and friends or keep as a resource for themselves. |

I-WOTCH, Improving the Wellbeing of people with Opioid Treated CHronic pain.

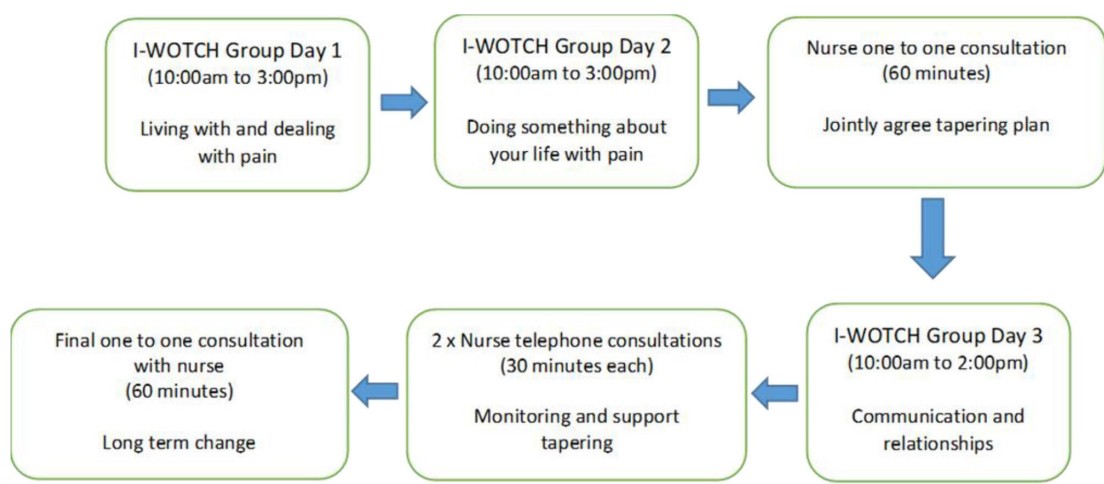

**Figure 3** Final model of Improving the Wellbeing of people with Opioid Treated CHronic pain (I-WOTCH) intervention.

the team developed a tapering App for use by the I-WOTCH trained nurses across sites. The I-WOTCH tapering App was developed by JN and SE working with the University of Warwick's Clinical Trials Unit (CTU) programming team (HA, CM and AW) and provided to the nurses on a handheld tablet. The I-WOTCH tapering App was based on a mathematical algorithm applying the Mayo Clinic tapering regime while accounting for UK commercial preparations. Nurses used the App to generate a participant specific tapering plan, which was synchronised to the I-WOTCH Trial database. The study team at Warwick CTU then logged into the centralised trial management website, printed and posted the tapering plan to the participant for their information and general practitioner for prescribing.

The I-WOTCH trained nurse entered the total daily dose of the participant-specific opioid preparation into the home screen of the App (eg, 60 mg oxycodone/day). The App algorithm then calculated 10% of the total daily dose and rounded this up or down to suit prescribing. All tablet, capsule or patch denominations of all opioid preparations were tabulated and added to the App to ensure the algorithm not only produced a 10% per week tapering regime but also recommended various prescribing methods (eg, oxycodone 35 mg could be prescribed as 30 and 5 mg or 20, 10 and 5 mg tablets or 10, 10, 10 and 5 mg tablets).

For patch preparations, we advised participants to taper using their original opioid if 10% was not achievable (eg, 12 µg of fentanyl being the smallest step down), the app algorithm was adjusted to recommend a 20% taper at 2-week intervals. Lowest dosage patch preparations were finally converted to slow release morphine equianalgesic doses and tapered accordingly.

### My opioid manager

The My Opioid Manager Book and App is the output of a project of Toronto Rehabilitation Institute, University Health Network. In 2010, Dr. Andrea Furlan, a Physician and Scientist at Toronto Rehabilitation Institute, developed a tool for physicians prescribing opioids for patients with chronic non-malignant pain. In 2012, the

Opioid Manager was converted to an App for smartphones and tablets. The My Opioid Manager Book (and App) is intended to complement the Opioid Manager by providing the same information in a format that can be used by people with chronic pain who are on opioids, or by people who are not on opioids but who might be considering this option to help manage their chronic pain. The goal of My Opioid Manager is preparing the patient for upcoming consultations with their healthcare provider. Some of the topics discussed include: understanding the causes of various types of pains, uses of opioids and the side effects and risks, managing pain by tracking opioid usefilness, and tips on using opioids. For this study, we Anglicised the content in terms of language used as well as the name of medication brands and pictures to be more representative of the UK population.

### Venue for delivering the intervention

Where possible, the I-WOTCH intervention is delivered in the community. Factors to consider when booking a venue included, access to a building, parking and public transport links, a room to accommodate participants and facilitators with chairs and equipment, stairs, lifts, kitchen facilities and room for equipment such as a flipchart, laptop screen, speakers and internet access.

### I-WOTCH facilitator training

The delivery–receipt–enactment chain of the I-WOTCH intervention provided a framework for training of facilitators and defining dosage received for participants to promote behaviour change (opioid tapering).[32] The I-WOTCH training included two full days for all facilitators (clinical and lay facilitators) and an additional day for clinical facilitators only, to learn the clinical aspects of tapering, opioid specific education, generating tapering plans and MI for the one-to-one consultations. The design of the training package and implementation was adapted to Kolb's experiential learning cycle (training, experience and reflective observation).[33] The training days gave all facilitators exposure to the different components of the intervention through education and use of case studies. Trainers were given copies of the

I-WOTCH manual and all participant intervention materials. Throughout the training days, facilitators had the opportunity to ask questions and get clarity on any of the topics being covered. At the end of the training, a short assessment was completed by each trainee. If any of the trainees scored below 70%, they were then contacted by phone to go over any areas needing further explanation and offered further training if needed.

## DISCUSSION

We have used a theory driven approach to developing an intervention for opioid reduction for people with chronic non-malignant pain. Based on the COPERS intervention for the management of pain, the best available empirical evidence at the time, and consultation with lay people, we have developed a manualised intervention and training package. It has been piloted, revised and adapted considering all feedback received. The I-WOTCH intervention has the potential to help people reduce their opioid use and improve their overall quality of life. We are not aware of any other programme of analogous interventions targeting similar populations. Previous non-pharmacological interventions have included mindfulness, cognitive–behavioural therapy and meditation and the use of electroacupuncture which showed no reduction in the number of participants who ceased their opioid use.[4] The I-WOTCH intervention differs in that it combines group and one to one support, with the mechanisms of change and opioid reduction targeted through peer support, education, case studies, reflection and MI. It is a time and resource intensive intervention, however, having a multicomponent intervention will increase the potential to address the complex psychological, social and physical aspects of opioid tapering. We have developed an opioid tapering App which can be used to calculate individual opioid tapering plans.

The roll out and scalability of the I-WOTCH training have been considered, a step-by-step manual with materials to set up and deliver the programme was created. The I-WOTCH facilitator training can be delivered to groups of clinicians and ongoing support given throughout the delivery of the intervention. The I-WOTCH trial will allow us to assess: the delivery of the intervention on a large scale, the training of multiple facilitators and managing the group element of the programme.

## CONCLUSION

We have designed an opioid reduction intervention package suitable for testing in a randomised controlled trial.

**Author affiliations**
[1]Warwick Clinical Trials Unit, Warwick Medical School, University of Warwick, Coventry, UK
[2]Department of Pain Medicine, James Cook University Hospital, Middlesbrough, UK
[3]Wolfson Institute of Population Health, Barts and The London School of Medicine and Dentistry, Queen Mary University of London, London, UK
[4]Toronto Rehabilitation Institute, University Health Network, Department of Medicine, University of Toronto, Toronto, Ontario, Canada
[5]Department of Medicine, Institute for Work & Health, Toronto, Ontario, Canada
[6]UNTRAP, University of Warwick, Coventry, UK
[7]SUCE, Coventry University, Coventry, UK
[8]Department of Psychology, University of Warwick, Coventry, UK
[9]University Hospitals Coventry and Warwickshire, Coventry, UK

**Acknowledgements** We would like to thank Ms Sally Brown for her valuable input in the development of the intervention. We would also like to thank all of our PPI volunteers and the Clinical Research Network (North East and Cumbria) for hosting the events which provided valuable insight to the structure and content of the intervention. We would like to thank our clinical and lay facilitators who attended training, delivered the intervention and provided valuable feedback. We would like to thank all participants involved in the pilot stages of the intervention. We would also like to thank Dr Alison Hipwell for her contribution to the intervention development and Dr Celia Bernstein and Dr Jodie Westhead for their contribution to the I-WOTCH study.

**Collaborators** I-WOTCH team: Sharisse Alleyne, University of Warwick, Charles Abraham, Deakin University, Shyam Balasubramanian, Department of Anaesthesia & Pain Medicine University Hospitals Coventry and Warwickshire NHS Trust, Lauren Betteley, University of Warwick, Katie Booth, University of Warwick, Kirstie Haywood, University of Warwick, Sheeja Manchira Krishnan, The University of York, Ranjit Lall, University of Warwick. Andrea Manca, The University of York. Dipesh Mistry, University of Warwick, Vivien P Nichols, University of Warwick, Emma Padfield, University of Warwick, Anisur Rahman, Kate Seers, University of Warwick, University College London, Cynthia Paola Iglesis Urrutia.

**Contributors** All authors read and approved the manuscript. HKS and SE are Co-Chief Investigators and oversee the running of the study. All named authors contributed to the design and/or delivery of the I-WOTCH intervention. HKS, SE, JS and DC were involved in the design of the I-WOTCH intervention and design and delivery of facilitator training. CT contributed to the design and delivery of the I-WOTCH intervention, providing feedback on all materials and a trained facilitator. ADF developed My Opioid Manager, the content was anglicised for this study and also contributed to the design of the overall I-WOTCH intervention. SE, JN, HA, CM and AW developed the I-WOTCH Opioid tapering App. MU, NKYT and ST contributed to the design of the intervention, training manuals and to this manuscript. HKS is responsible for the overall content as guarantor.

**Funding** This project has received funding from the National Institute for Health Research (NIHR), Health Technology Assessment (HTA) (project number 14/224/04). SJCT supported by the National Institute for Health Research ARC North Thames.

**Disclaimer** The views and opinions expressed therein are those of the authors and do not necessarily reflect those of the HTA, NIHR, NHS or the Department of Health.

**Competing interests** SE is the Chair of the specialised pain CRG at NHS England, he is Chief investigator and principal investigator of a number of NIHR and Industry funded trials, he has received personal fees from Medtronic Ltd, Mainstay Medical, Boston Scientific Corp for consultancy work. His department has received research funding from the National Institute of Health Research, Medtronic Ltd and Boston Scientific Corp. HS is director of Health Psychology Services Ltd, providing psychological services for a range of health related conditions. NKYT is chief investigator or coinvestigator of other chronic pain related projects funded by the NIHR, MRC, Warwick-Wellcome Translational Partnership. MU is chief investigator or coinvestigator on multiple previous and current research grants from the UK National Institute for Health Research, Arthritis Research UK and is a coinvestigator on grants funded by the Australian NHMRC. He was an NIHR Senior Investigator until March 2021. He has received travel expenses for speaking at conferences from the professional organisations hosting the conferences. He is a director and shareholder of Clinvivo Ltd that provides electronic data collection for health services research. He is part of an academic partnership with Serco Ltd, funded by the European Social Fund, related to return to work initiatives. He receives some salary support from University Hospitals Coventry and Warwickshire. He is a coinvestigator on three NIHR funded studies receiving additional support from Stryker Ltd. He has accepted honoraria for teaching/lecturing from consortium for advanced research training in Africa. Until March 2020, he was an editor of the NIHR journal series, and a member of the NIHR Journal Editors Group, for which he received a fee. ADF is author of the My Opioid Manager book and App distributed in iTunes and Google Play. Both book and app are free of charge. She is author of the Opioid Manager App, a paid app distributed only in iTunes for healthcare professionals. The app is owned by UHN, the hospital where ADF works. ADF does not get any financial benefit from the sales of the app. ADF has a monetized YouTube channel since January 2021 that contains some videos about opioids and opioid tapering. Since April 2021, ADF has an unrestricted educational grant

to maintain an online self-assessment opioid course for healthcare professionals in Canada. The funding is provided by the Canadian Generics Pharmaceutical Association (CGPA). The funding organisation has no role in the preparation, approval, recruitment of participants, or data analysis of the course content. Responsibility for the course content is solely that of the authors. ST is chief investigator or coinvestigator on multiple previous and current research grants from the UK National Institute for Health Research.

**Patient and public involvement**  Patients and/or the public were involved in the design, or conduct, or reporting, or dissemination plans of this research. Refer to the Methods section for further details.

**Patient consent for publication**  Not applicable.

**Ethics approval**  Ethics approval was given by Yorkshire & The Humber - South Yorkshire Research Ethics Committee on 13 September 2016 16/YH/0325. Participants gave informed consent to participate in the study before taking part.

**Provenance and peer review**  Not commissioned; externally peer reviewed.

**Data availability statement**  No data are available. No additional data available.

**ORCID iDs**
Harbinder Kaur Sandhu http://orcid.org/0000-0003-1522-8078
Andrea D Furlan http://orcid.org/0000-0001-6138-8510
Nicole K Y Tang http://orcid.org/0000-0001-7836-9965
Stephanie JC Taylor http://orcid.org/0000-0001-7454-6354
Martin Underwood http://orcid.org/0000-0002-0309-1708

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
