## [Reviewer comments · BMJ Open]

ARTICLE DETAILS

TITLE (PROVISIONAL)	The development and testing of an opioid tapering self-management intervention for chronic pain – I-WOTCH
AUTHORS	Sandhu, Harbinder; Shaw, Jane; Carnes, Dawn; Furlan, Andrea; Tysall, Colin; Adjei, Henry; Muthiah, Chockalingam; Noyes, Jennifer; Tang, Nicole; Taylor, Stephanie; Underwood, Martin; Willis, Adrian; Eldabe, Sam

VERSION 1 – REVIEW

REVIEWER	Frank, Bernhard The Walton Centre NHS Foundation Trust We also just received funding for a feasibility study using a brief psychological intervention to help patients weaning of opioids. We also submitted a research proposal for the HTA call funding this study.
REVIEW RETURNED	02-Jul-2021

GENERAL COMMENTS	The authors describe the process of developing the content of educational group and one to one session being used to help patient reducing or cease the opioids used for treating non-malignant pain. The different topics covered in the educational sessions and the format were produced during patient group meetings. The details of the content was mapped in a table to aims, theoretical underpinning and what behavioural change was targeted according a recognised taxonomy. The resulting intervention was then tested in two feasibility stages the second one being the pilot phase of a large RCT. The result of this exercise was the final i-Wotch intervention used in the RCT. The choice and rational behind the digital technology used is well described in the paper. I belief that one should be able to replicate the process described in most health care settings in the UK to translate the i-wotch intervention into a opioid tapering programme delivered in the community. Reference 8 and 9 are identical. There are some small typos. e.g.: page 6 line 50 were: instead of were to: This paper provides very nice insight how to develop a complex intervention with patient input that can subsequently be piloted and tested in an RCT.
--

REVIEWER	Geraghty, Adam University of Southampton, Primary Care and Population Sciences
REVIEW RETURNED	17-Nov-2021

GENERAL COMMENTS	Thank you for the opportunity to review this interesting development paper. It is a well written description of the intervention development process; therefore my comments will be brief: 1) In the introduction there is mention of systematic review of opioid deprescribing interventions. This seems worthy of more attention than is given to it currently. I think the description of this review should be expanded. OK, the authors did not recommend one regime, but what were the effectiveness findings? Do these things work? What was the format of the interventions generally? Did this influence the design of the intervention to be prescribed? 2) With regard to supplemental materials, a CD and DVD is mentioned. Do people still have widespread access to DVD and CD players? I would have thought a website containing these materials would have been more appropriate. 3) Whilst the method is well documented, it seems a lot of weight has been placed on the PPI feedback groups, no doubt this is important, but I wondered if the team had considered conducting structured, systematic qualitative work? Rather than PPI discussions? And if not why not? The same point for the evaluation in the pilot? 4) In the discussion it would be useful if the authors could consider the potential issues of this design for rollout and implementation? There is a fair amount of training needed for staff, and commitments for group and individual sessions for patients. How scalable and accessible is this design?
--

VERSION 1 – AUTHOR RESPONSE

Reviewer: 1	
Reference 8 and 9 are identical	Thank you the references have been amended.
There are some small typos. e.g.: page 6 line 50 were: instead of were to:	Thank you this has now been corrected.
Reviewer: 2	
In the introduction there is mention of systematic review of opioid deprescribing interventions. This seems worthy of more attention than is given to it currently. I think the description of this review should be expanded. OK, the authors did not recommend one regime, but what were the effectiveness findings? Do these things work? What was the format of the interventions generally? Did this influence the design of the intervention to be prescribed?	Thank you we have now included more detail on the types of interventions and effectiveness which further strengthens the need for the development and testing of I-WOTCH.
With regard to supplemental materials, a CD and DVD is mentioned. Do people still have widespread access to DVD and CD players? I would have thought a website containing these materials would have been more appropriate.	Thank you and yes, we agree. However, at the time the intervention was initially developed this was not the case. We have now made this available as an on-line resource.
Whilst the method is well documented, it seems a lot of weight has been placed on the PPI feedback groups, no doubt this is	Thank you, the role of PPI input into the joint development and evaluation of interventions is very different from using qualitative data to address

important, but I wondered if the team had considered conducting structured, systematic qualitative work? Rather than PPI discussions? And if not why not? The same point for the evaluation in the pilot?	specific research questions. Using a formal research framework would change our patient partners from colleagues working on the co-production of the intervention to being research subjects. We did however use qualitative research and a tested intervention for pain management “COPERS” to inform and develop the I-WOTCH intervention and we felt at this stage PPI was more important to address in this population.
In the discussion it would be useful if the authors could consider the potential issues of this design for rollout and implementation? There is a fair amount of training needed for staff, and commitments for group and individual sessions for patients. How scalable and accessible is this design?	Thank you, we have now addressed this in the discussion. We have outlined the importance of the trial to test the delivery of the intervention training and the actual intervention on large scale across multiple sites. The design of the intervention is scalable and accessible, with a fully manualised practitioner step by step guide and an intense three-day training which also uses case studies to aid learning. Multiple facilitators can be trained together.

VERSION 2 – REVIEW

REVIEWER	Geraghty, Adam University of Southampton, Primary Care and Population Sciences
REVIEW RETURNED	16-Dec-2021
GENERAL COMMENTS	The authors have responded well, addressing my comments.